# A High-Precision Current-Mode Bandgap Reference with Nonlinear Temperature Compensation

**DOI:** 10.3390/mi14071420

**Published:** 2023-07-14

**Authors:** Zhizhi Chen, Qian Wang, Xi Li, Sannian Song, Houpeng Chen, Zhitang Song

**Affiliations:** 1State Key Laboratory of Materials for Integrated Circuits, Shanghai Institute of Microsystem and Information Technology, Chinese Academy of Sciences, 865 Changning Road, Shanghai 200050, China; 2Schools of Microelectronics, University of Chinese Academy of Sciences, Beijing 100049, China

**Keywords:** bandgap current reference, high-order curvature-compensated technique, temperature coefficient (TC), current-mode reference

## Abstract

A high-precision current-mode bandgap reference (BGR) circuit with a high-order temperature compensation is presented in this paper. In order to achieve a high-precision BGR circuit, the equation of the nonlinear current has been modified and the high-order term of the current flowing into the nonlinear compensation bipolar junction transistor (NLCBJT) is compensated further. According to the modified equation, two solutions are designed to improve the output accuracy of BGR circuits. The first solution is to divide the NLCBJT branch into two branches to reduce the coefficient of the nonlinear temperature compensation current. The second solution is to inject the nonlinear current into the two branches based on the first one to further eliminate the temperature coefficient (TC) of the current flowing into the NLCBJT. The proposed BGR circuit has been designed using the Semiconductor Manufacturing International Corporation (SMIC) 55 nm CMOS process. The simulation results show that the variations in currents flowing into NLCBJTs improved from 148.41 nA to 69.35 nA and 7.4 nA, respectively, the TC of the output reference current of the proposed circuit is approximately 3.78 ppm/°C at a temperature range of −50 °C to 120 °C with a supply voltage of 3.3 V, the quiescent current consumption of the entire BGR circuit is 42.13 μA, and the size of the BGR layout is 0.044 mm^2^, leading to the development of a high-precision BGR circuit.

## 1. Introduction

Bandgap reference (BGR) circuits are critical modules in most integrated circuit systems and are widely used in analog circuits, digital circuits, and mixed-signal circuits, such as memory circuits, A/D converters, and low dropout linear regulators. BGR circuits provide temperature-independent voltages or currents for the system-on-a-chip (SoC), and their performance determines the quality of the entire SoC. With the development of the CMOS process, the feature size of integrated circuits continues to decrease, and the operation voltage of the electronic system is becoming increasingly lower. Low-voltage and high-precision BGR circuits have received widespread attention.

The output voltage of conventional voltage-mode BGR circuits with first-order temperature compensation is 1.25 V approximately, which can achieve a TC of about a few tens of ppm/℃. In order to achieve more accurate reference voltages, higher-order temperature-compensated techniques are required for BGR circuits. Rincon-Mora et al. [1] adjusted the reference voltage by optimizing the temperature component with the trimming process and achieved high accuracy of the output voltage. Leung et al. [2] proposed that the ratio of resistors with the same type and size is independent of temperature, which can be used to reduce temperature drift. Ker et al. [3] used a subtraction circuit to cancel the convex curve or the concave curve of the output reference current of two BGR circuits. Exponential temperature compensation [4], quadratic temperature compensation [5], and third-order compensation [6] were also used to cancel the high-order terms of the emitter-base voltage V_EB_, eliminate the temperature drift, and obtain a reference voltage with a very small TC.

In modern CMOS technology, the operation voltage of CMOS devices is lower than 1.2 V, so a reference voltage should be lower than 1.2 V. Banba et al. [7] proposed a sub-1V BGR circuit in which the current-mode technique is adopted to scale down the output reference voltage, and a variety of high-precision BGR circuits are developed.

Compared to the first-order temperature compensation in [7], Malcovati [8] introduced a high-order temperature compensation, which is based on the theory that the current in the nonlinear compensation bipolar junction transistor (NLCBJT) is temperature-independent. However, high-order temperature residue terms still exist in the NLCBJT currents in this circuit, which require further rejection or elimination.

In this paper, the accuracy of the BGR is further improved on the basis of [8]. The rest of this paper is organized as follows: Section 2 describes the operation principle of a conventional current-mode BGR circuit; Section 3 describes the two proposed solutions for nonlinear compensation BGR circuits; Section 4 presents the simulation results that verify the accuracy of the proposed high-order terms compensated circuit; and the conclusions are provided in Section 5.

## 2. Principle of Conventional Current-Mode BGR Circuits

The low-voltage BGR circuit proposed by Banba et al. [7] is a current-mode BGR circuit whose output reference current *I_REF_* is realized by the sum of two currents. One is complementary to the absolute temperature *I_CTAT_*, and the other is proportional to the absolute temperature *I_PTAT_*. First, a temperature-independent reference current was generated.

As presented in Figure 1, due to the effect of negative feedback, the relationship *V_A_* = *V_B_* = *V_EB_*_1_ and a PTAT current *I_PTAT_* proportional to *V_T_* is achieved. With additional equal resistors *R*_1_ and *R*_2_ (*R*_1_ = *R*_2_ = *R*_1,2_), the BGR circuit achieves a CTAT current *I_CTAT_* proportional to *V_EB_*. The currents *I*_1_ and *I*_2_ are the sum of the currents *I_PTAT_* and *I_CTAT_*, flowing through the current mirror that consists of the transistors M_0_, M_1,_ and M_2_ with the same aspect ratio. The currents can be expressed as follows:(1)$$I_{1}=I_{2}=I_{REF}=\frac{v_{T}\ln N}{R_{0}}+\frac{V_{EB1}}{R_{1,2}}=\frac{kT}{qR_{0}}\ln N+\frac{V_{EB1}}{R_{1,2}}$$

Then, a low reference voltage *V_REF_* can be generated and expressed as
(2)$$V_{REF}=I_{REF}R_{3}=I_{1}R_{3}=I_{2}R_{3}=\left(\frac{∆V_{EB}}{R_{0}}+\frac{V_{EB1}}{R_{1,2}}\right)R_{3}$$

However, the current-mode BGR circuit shown in Figure 1 still belongs to first-order temperature compensation. A large high-order temperature current flows into the emitter of *Q*_1_, which affects the accuracy of *V_REF_*. According to the study by Tsividis et al. [9], an accurate analysis of the temperature effects on *V_EB_*-*T* characteristics can be expressed as
(3)$$V_{EB}\left(T\right)=V_{G0}\left(T_{r}\right)+\left(\frac{T}{T_{r}}\right)\left[V_{EB}\left(T_{r}\right)-V_{G0}\left(T_{r}\right)\right]-\left(n-\delta \right)\frac{kT}{q}ln\left(\frac{T}{T_{r}}\right)$$
where *V_G_*_0_(*T_r_*) is the bandgap voltage of silicon at the reference temperature *T_r_*, n is a temperature-independent and process-dependent constant around 4, and *δ* is a factor of the temperature dependent on the collector current, which is equal to 1 if the current in the BJT is PTAT and becomes 0 when the current is temperature-independent. *V_T_* is the thermal voltage, *k* is Boltzmann’s constant, and *q* is the electric charge.

In Equation (3), the second item has a first-order TC, whereas the third item is a high-order temperature nonlinear term that should be rejected or eliminated to achieve a high-precision BGR.

On the basis of [7], Malcovati et al. [8] presented a high-precision BGR circuit with a low supply voltage, where nonlinear currents were generated to compensate for the high-order errors, as shown in Figure 2.

The currents flowing into *Q*_0_ and *Q*_1_ are proportional to the absolute temperature so that the parameter *δ* in the expression of *V_EB_* is equal to 1. Since the currents flowing into Q_2_ are temperature-independent, the parameter *δ* is equal to 0 [8].

The *V_EB_* of *Q*_0_ and *Q*_1_ can be expressed as
(4)$$V_{EB,Q0,1}=V_{G0}\left(1-\frac{T}{T_{r}}\right)+V_{EB0}\left(\frac{T}{T_{r}}\right)-\left(n-1\right)\frac{kT}{q}ln\left(\frac{T}{T_{r}}\right)$$

The current in M_0_ is
(5)$$I_{REF}=I_{PTAT}+I_{CTAT}-I_{NL}$$
which is the current with a low TC after high-order temperature nonlinear compensation. The current is copied by M_2_ and injected into a diode connected to NLCBJT *Q*_2_, which is expressed by
(6)$$I_{Q2}=I_{PTAT}+I_{CTAT}-3I_{NL}$$

Because the nonlinear current I_NL_ is very small, its TC can be ignored [8]. Then, a *V_EB_* with *δ* = 0 is produced across *Q*_2_, which can be expressed as
(7)$$V_{EB,Q2}=V_{G0,Q2}\left(1-\frac{T}{T_{r}}\right)+V_{EB0,Q2}\left(\frac{T}{T_{r}}\right)-n\frac{kT}{q}ln\left(\frac{T}{T_{r}}\right)$$

Equation (7) is subtracted from (4) and leads to a nonlinear voltage *V_NL_*, which is expressed as
(8)$$V_{NL}=V_{EB,Q2}-V_{EB,Q0,1}=-\frac{kT}{q}ln\left(\frac{T}{T_{r}}\right)+\Delta V_{EB,Q2,Q1}\left(\frac{T}{T_{r}}\right)$$
where the first term in the equation is the nonlinear term of temperature, and the second term is the error of the linear term of the *V_EB_* of two BJT *Q*_1_ and *Q*_2_ with the same geometry; however, the emitter currents *I_E,Q_*_1_ and *I_E,Q_*_2_ are not equal, so the error of the linear term is not equal to zero. Equation (8) is then corrected.

The values of resistors R_3_ and R_4_ are equal, and the *I_NL_*s generated on them are equal. Then, the current of *I_PTAT_* + *I_CTAT_* − *I_NL_* follows M_4_ and R_5_, as shown in Figure 3.

The output reference voltage *V_REF_* becomes
(9)$$V_{REF}=V_{T}\frac{R_{5}ln\left(N\right)}{R_{0}}+V_{EB,Q0,1}\frac{R_{5}}{R_{1,2}}+V_{NL}\frac{R_{5}}{R_{3,4}}=\frac{R_{5}}{R_{1,2}}\left(\frac{R_{1,2}ln\left(N\right)}{R_{0}}V_{T}+V_{EB,Q0,1}-\frac{R_{1,2}}{R_{3,4}}V_{NL}\right)$$
where the third term is the nonlinear part, and it can be used to effectively compensate for the nonlinear item of the second term, *V_EB,Q_*_0,1_. By substituting (4) into (9) and setting n − 1 equal to $\frac{R_{1,2}}{R_{3,4}}$, the nonlinear temperature term in *V_REF_* can be eliminated, and a high-precision reference voltage can be achieved.

This BGR circuit achieves an output reference voltage of 0.536 V and obtains a TC of 7.5 ppm/K over a wide temperature range of 80 °C (from 0 °C to 80 °C). Compared to the BGR circuit without the curvature correction technique, the BGR circuit is improved by about three times.

However, in this structure, the expression for the emitter current of *Q*_2_ is actually
(10)$$I_{E,Q2}=I_{PTAT}+I_{CTAT}-3I_{NL}$$
where there are excess high-order temperature terms with a certain impact on *I_REF_*. The coefficient of this current can be further reduced, and a more accurate *I_REF_* can be achieved.

## 3. Proposed High-Precision Current-Mode BGR Circuit

In order to completely eliminate high-order temperature terms in the current flowing through *Q*_2_, a novel high-precision compensation BGR structure is proposed in this paper.

On the basis of the conventional curvature-compensated BGR circuit, the transistor M_3_ is first added to mirror the current in M_0_ flowing into BJT *Q*_3_ and then form a new branch, which can share the nonlinear current flowing into the same BJT, as shown in Figure 3.

At this time, the current flowing through *Q*_2_ and *Q*_3_ is
(11)$$I_{E,Q2,3}=I_{PTAT}+I_{CTAT}-2I_{NL}$$

The TC of the nonlinear compensation BJT current is reduced, which makes the output reference current more accurate.

In order to further eliminate high-order temperature terms, a BGR circuit with high-order temperature compensation was designed, as shown in Figure 4.

Let R_5_ be equal to R_3_ and R_4_ (R_3_ = R_4_ = R_5_). Based on the characteristics of the operational amplifier (OPAMP) A_1_ and A_2_, two branches, each with a nonlinear current similar to I_NL_, are formed and injected into Q_2_ and Q_3_ to offset the excess high-order temperature terms.

The circuit of the OPAMP A_0_, A_1,_ and A_2_ is provided in Figure 5, where the input stage of this circuit mainly consists of a PMOS transistor differential pair M_21_ and M_22_ and an NMOS transistor differential pair M_23_ and M_24_ placed in parallel as a rail-to-rail differential input stage, whose range of input common-mode voltage can be from ground to V_DD_. The dominant pole of the circuit is located at the output port. The product of the equivalent impedance and capacitance is large, so the position of the pole is close to the DC point. And the non-dominant pole of the circuit is located at the node between the drain of M_32_ and the source of M_33_, and the other pole is located between the source of M_34_ and the drain of M_35_. The output impedance and parasitic capacitance of these two nodes are both small so that their poles are far from the dominant pole. So this circuit can be regarded as only one pole approximately and is kept stable through simple compensation.

Finally, the current is
(12)$$I_{E,Q2,3}=I_{PTAT}+I_{CTAT}-I_{NL}$$

This formation is theoretically completely independent of temperature.

The design details of the proposed high-precision current-mode BGR and OPAMP circuits A_0_, A_1_, and A_2_ are provided in Table 1.

With the same power supply voltage, the same component sizes, and the same temperature range from −50 °C to 120 °C, the current *I_PTAT_* + *I_CTAT_* − 3*I_NL_* flowing into Q_2_ in Figure 2, the current *I_PTAT_* + *I_CTAT_* − 2*I_NL_* flowing into Q_2_ or Q_3_ in Figure 3, and the current *I_PTAT_* + *I_CTAT_* − *I_NL_* flowing into Q_2_ or Q_3_ in Figure 4 simulated in the SMIC 55 nm CMOS process are shown in Figure 6.

Compared to Figure 2 and Figure 3, the current curve in Figure 4 is the most stable, and the variation is the smallest. In other words, the temperature stability is the best.

And the output reference currents I_REF_ of the structures in Figure 2, Figure 3 and Figure 4 simulated in the SMIC 55 nm CMOS process are shown in Figure 7.

The I_REF_ of the circuit in Figure 2 varies from the minimum of 4.2049 μA to the maximum of 4.2227 μA with a change of 17.8 nA, the I_REF_ in Figure 3 varies from the minimum of 4.1817 μA to the maximum of 4.1926 μA with a change of 10.9 nA, and the I_REF_ in Figure 4 varies from the minimum of 4.1539 μA to the maximum of 4.1563 μA with a change of 2.4 nA. It can be seen that the I_REF_ of the proposed high-precision current-mode BGR circuit is more stable significantly.

## 4. Simulation Results

The proposed current-mode BGR circuit with a high-order temperature compensation was designed using the SMIC 55 nm CMOS process. The size of the layout of the proposed circuit including dummies turned out to be 300.43 μm × 148.67 μm, which is illustrated in Figure 8.

A.The output reference current

With a supply voltage of 3.3 V, the I_REF_ of the proposed circuit measured from −50 °C to 120 °C is presented in Figure 9.

The equation of TC can be expressed as
(13)$$TC=\frac{V_{REF,max}-V_{REF,min}}{V_{REF,ave}\times \left(T_{max}-T_{min}\right)}\times 10^{6}$$

In the current-mode BGR circuit, the *V_REF_* in the above equation should be replaced by *I_REF_*, while the rest remains unchanged. *I_REF_* over the whole temperature range is about 2.4 nA, varying from 3.6829 μA to 3.6853 μA. So, the typical TC can be calculated as 3.78 ppm/°C.

B.Monte Carlo simulation

The Monte Carlo simulation is conducted to assess the circuit stability due to the influence of the process and mismatched variations. Three hundred iterations of the generated I_REF_ are shown in Figure 10.

The simulation results show that I_REF_ varied from 3.6789 μA to 3.6881 μA under the worst-case scenario in Figure 10a, whose TC is about 7.64 ppm/°C. And it can be calculated that the mean value μ of TC is 4.51 ppm/℃, and the mean square error σ is 0.615 ppm/°C in Figure 10b.

The Monte Carlo simulation covers over 95% of the process corners and mismatches, which ensures a certain qualification rate for the product. However, the process corners have significant process variations under extreme conditions and require to be trimmed. Under the process corner of ss, there is a maximum deviation of 6 nA from the typical value in this paper. Due to the accuracy requirements of the BGR circuit, trims need to be made. Three-bit trimming is adopted, which means that there are eight trimming states. Three states greater than the typical value are set, and each state can be stepped by 3 nA, so a total of 9 nA can be stepped. Four trimming states below the typical value are set, and each state can be stepped by 3 nA, and a total of 12 nA can be stepped. Because trimming is an engineering implementation process, this paper does not provide detailed circuit implementation steps.

Figure 11 shows the simulation result of the output I_REF_ versus the temperature of the process corners, including ff, fs, sf, and ss, where the process corners of ss and fs are trimmed in one step to obtain better results, and those of ff and sf are maintained in the set state of tt without any trimming. At the worst process corner ff, the TC is about 7.97 ppm/°C.

C.Stability

Figure 12 shows the AC analysis results of the BGR for the gain and phase frequency response of the process corners, including tt, ff, fs, sf, and ss of the proposed circuit.

It can be observed that the phase margin is better than 76.63 degrees, and the gain margin is about 16.64 dB. When the gain is 0 dB, the phase margin is much greater than 60 degrees, which is very stable.

D.Transient response

Figure 13 illustrates the start-up process of the proposed circuit with a supply voltage V_DD_ step from 0 V to 3.3 V at an edge time of 1 ms, and when I_REF_ flows through a high-precision resistor with a temperature compensation of 160 kΩ, the proposed BGR circuit takes 0.14 ms to reach the normal operating state.

E.Comparison between the simulated characteristics of the proposed design and other works

The performance of the proposed BGR circuit is compared to that of other previous BGR circuits [10,11,12,13,14], as shown in Table 2.

It can be observed from Table 2 that due to more accurate high-order compensation, the TC of the proposed structure is superior to that of previous works over a wider range of temperatures, and there are also certain comprehensive advantages in layout area, current consumption, and PSR.

In order to evaluate the overall performance of the BGR circuits, an evaluation parameter figure-of-merit (*FOM*) defined in this paper can be expressed as
(14)$$FOM=\frac{\left|PSR\right|\times Temp\;range}{Best\;TC\times I_{Q}}$$

Because the layout area is related to the process, for the purpose of a fair comparison, the parameter of the layout area is not used in *FOM*, which only uses the temperature range, best *TC*, quiescent current, and *PSR*. It can be seen from the results that the value of *FOM* in this paper is much higher than that in [11,15], almost similar to that in [12,13], but lower than that in [10]. However, the *FOM* of [10] is only because of the excellent parameter of *I_Q_*, while the rest of the performance is ordinary. The most important parameter of the BGR circuit is the temperature coefficient, which is best identified in this paper.

F.Post-layout simulation

The results of the post-layout simulations are shown in Table 3.

## 5. Conclusions

In this study, the equation of the nonlinear current was modified, the high-order term of the current flowing into the NLCBJT was compensated further, and a high-precision current-mode BGR circuit with a high-order temperature compensation was designed and simulated using Cadence SPECTRE with a SMIC CMOS 55 nm process. The simulation results verify that the output reference current has good temperature independence (TC ≈ 3.78 ppm/°C) with a supply voltage of 3.3 V, a better layout area, and power supply rejection ability (PSR ≈ −63.1 dB at 100 frequency). These results display an effective enhancement in the performance of the BGR circuit.

## Figures and Tables

**Figure 1 micromachines-14-01420-f001:**
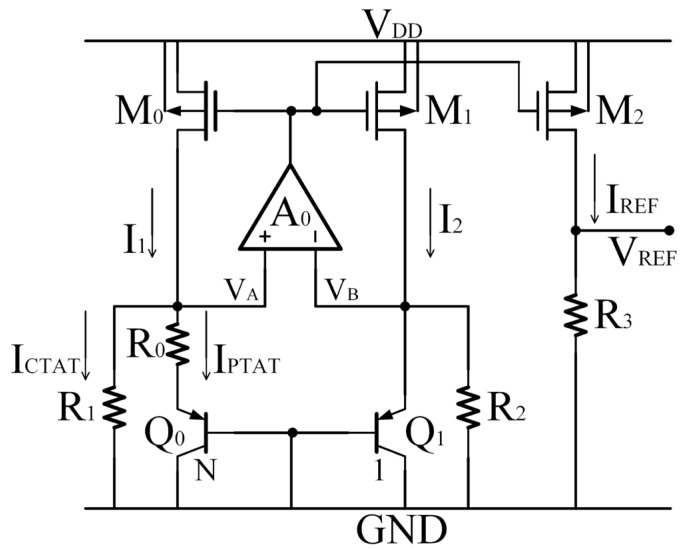
The BGR circuit proposed by Banba et al. [7].

**Figure 2 micromachines-14-01420-f002:**
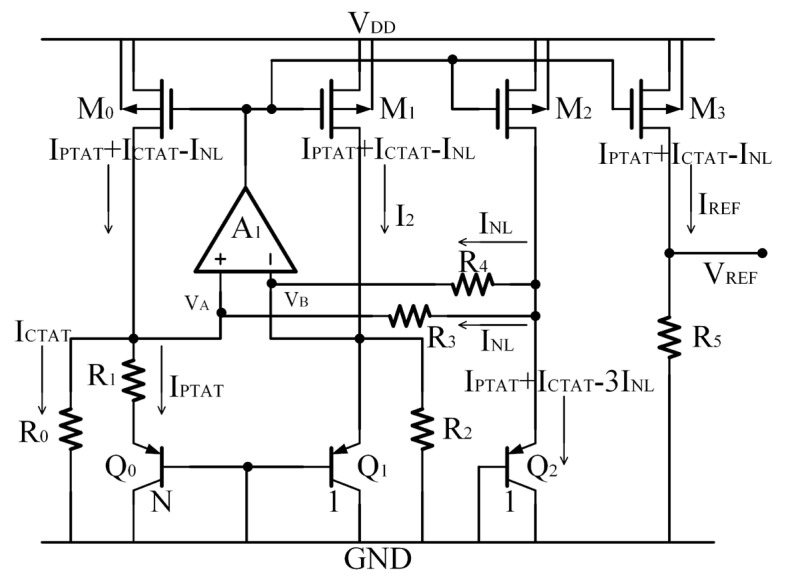
The high-accuracy BGR circuit proposed by Malcovati [8].

**Figure 3 micromachines-14-01420-f003:**
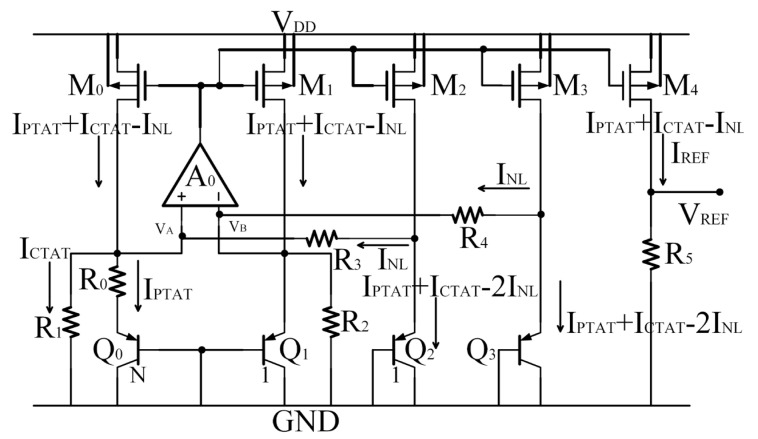
A solution for reducing the TC of current in NLCBJT.

**Figure 4 micromachines-14-01420-f004:**
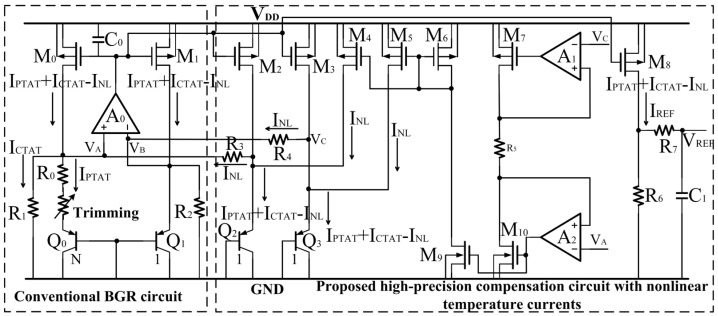
Proposed high-precision current-mode BGR circuit.

**Figure 5 micromachines-14-01420-f005:**
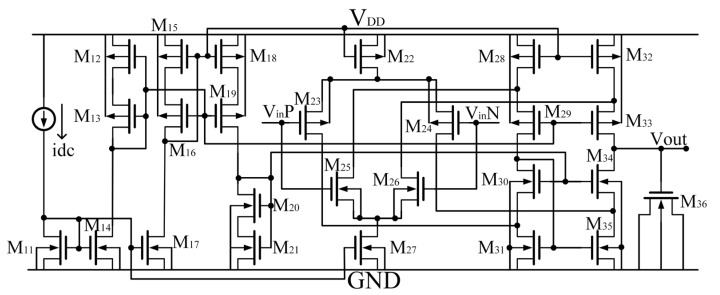
The OPAMP A_0_, A_1,_ and A_2_ of the circuit shown in Figure 4.

**Figure 6 micromachines-14-01420-f006:**
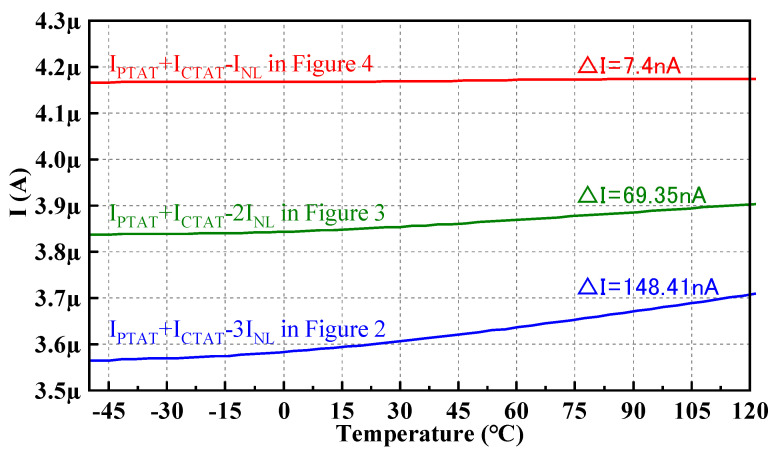
Currents flowing into Q_2_ or Q_3_ of the structures shown in Figure 2, Figure 3 and Figure 4.

**Figure 7 micromachines-14-01420-f007:**
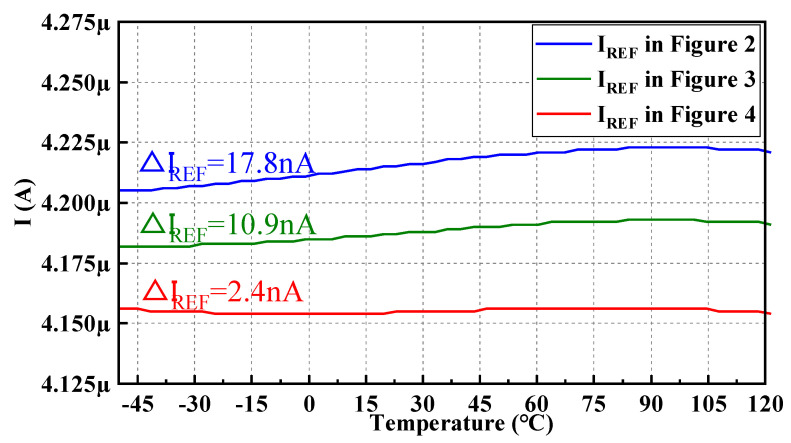
Measured I_REF_ of the structures shown in Figure 2, Figure 3 and Figure 4.

**Figure 8 micromachines-14-01420-f008:**
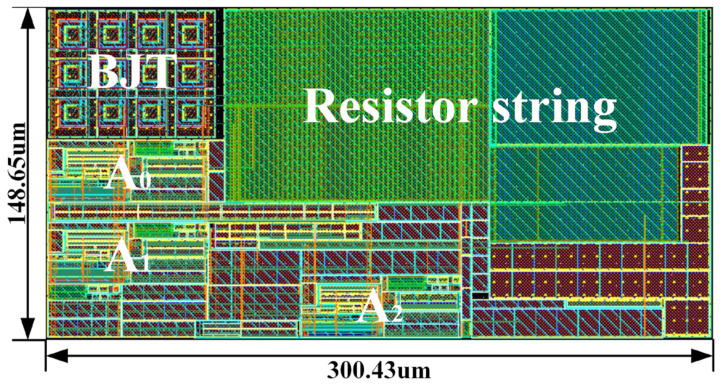
The layout of the proposed high-precision current-mode BGR circuit.

**Figure 9 micromachines-14-01420-f009:**
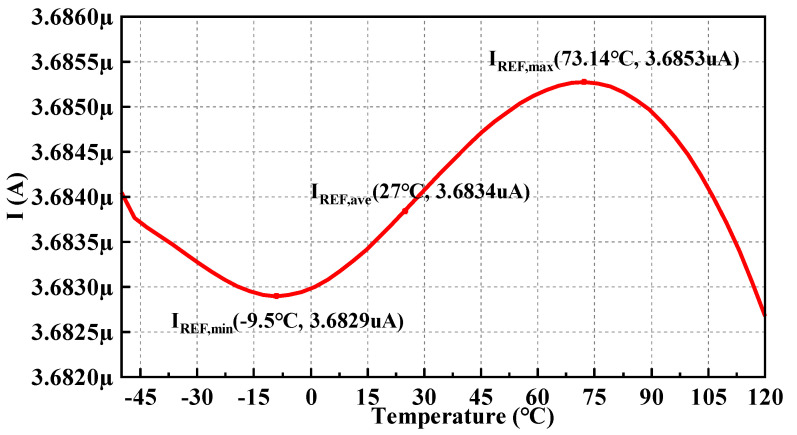
I_REF_ with temperature sweep at the process corner tt.

**Figure 10 micromachines-14-01420-f010:**
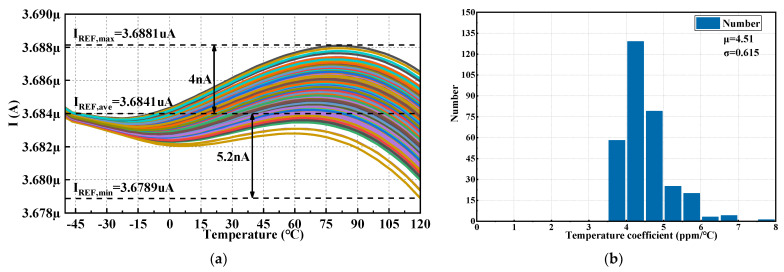
Monte Carlo simulation (300 iterations) for mismatch and process variations: (**a**) I_REF_ across temperature; (**b**) Temperature coefficient.

**Figure 11 micromachines-14-01420-f011:**
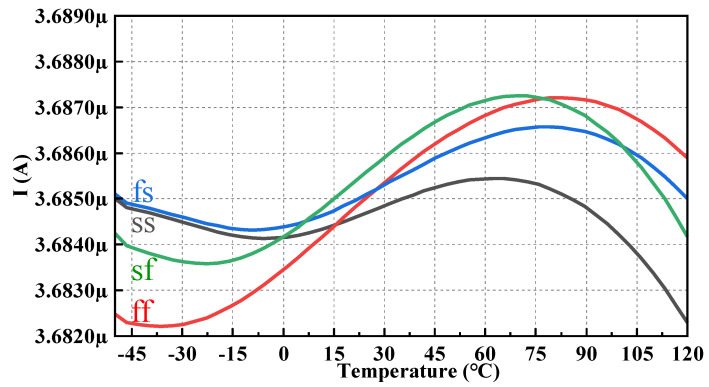
I_REF_ with temperature sweep of process corners including ff, fs, sf, and ss.

**Figure 12 micromachines-14-01420-f012:**
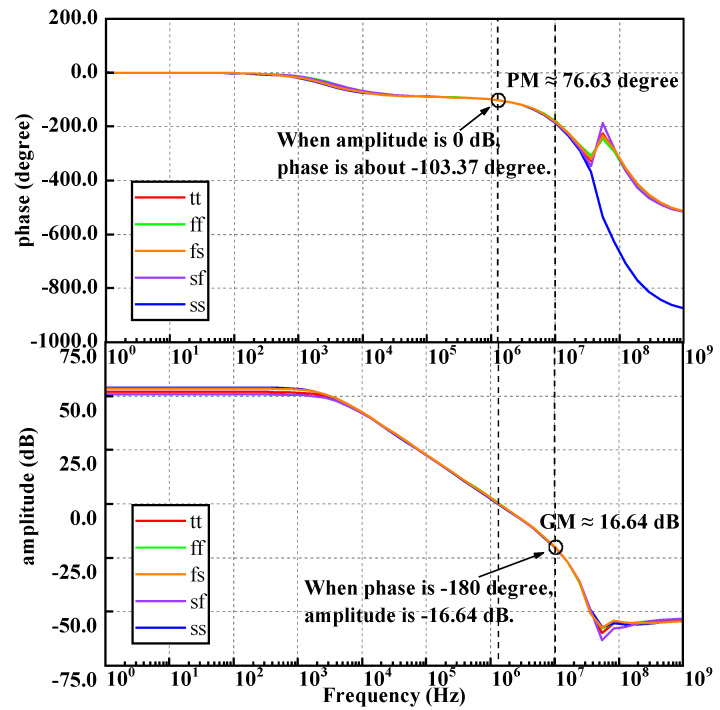
The gain and phase frequency response of every process corner.

**Figure 13 micromachines-14-01420-f013:**
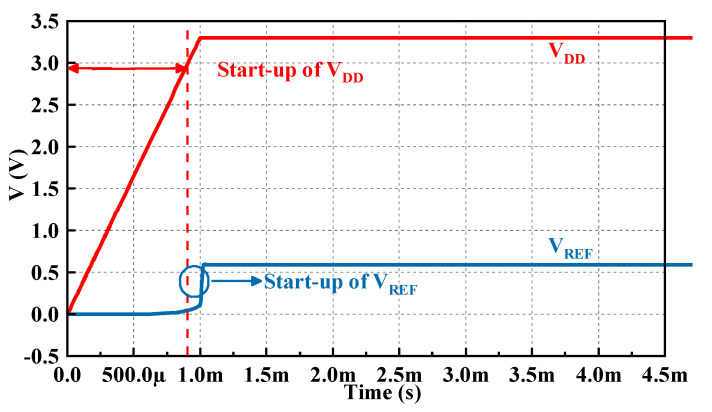
The transient response of the start−up process.

**Table 1 micromachines-14-01420-t001:** Component sizes used in the proposed BGR circuit.

Component	Parameter
M_0_, M_1_, M_2_, M_3_ and M_8_	W = 9 μm, L = 6 μm
M_4_, M_5_ and M_6_	W = 6 μm, L = 6 μm
M_7_	W = 3 μm, L = 6 μm
M_9_ and M_10_	W = 3 μm, L = 8 μm
M_11_, M_14_ and M_17_	W = 1 μm, L = 12 μm
M_12_	W = 1.8 μm, L = 6 μm
M_13_, M_16_, M_19_, M_29_ and M_33_	W = 1.5 μm, L = 6 μm
M_15_ and M_18_	W = 1.5 μm, L = 12 μm
M_22_, M_28_ and M_32_	W = 1.5 μm, L = 12 μm, m = 2
M_23_ and M_24_	W = 3 μm, L = 6 μm, m = 4
M_25_ and M_26_	W = 5 μm, L = 2.6 μm, m = 4
M_20_	W = 1.5 μm, L = 8 μm
M_21_, M_30_ and M_34_	W = 1 μm, L = 12 μm
M_27_, M_31_ and M_35_	W = 1 μm, L = 12 μm, m = 2
Q_0_	8 × (5.6 μm × 5.6 μm)
Q_1_, Q_2_ and Q_3_	1 × (5.6 μm × 5.6 μm)
R_0_	31.47 kΩ
R_1_ and R_2_	249.52 kΩ
R_3_, R_4_ and R_5_	62.38 kΩ
R_6_	160 kΩ

**Table 2 micromachines-14-01420-t002:** Performance summary and comparisons with other previous studies.

Parameters	This Work	Ref. [10]	Ref. [11]	Ref. [12]	Ref. [13]	Ref. [14]	Ref. [15]
Year	2023	2021	2019	2018	2012	2012	2010
Process	CMOS 55 nm	CMOS 65 nm	CMOS 0.18 μm	CMOS 0.5 μm	BiCMOS 0.5 μm	CMOS 0.35 μm	CMOS 0.5 μm
Supply voltage (V)	3.3	1.0–1.4	3.5–5	2.1–5	3.6	2.5	3.6
Layout area (mm^2^)	0.044	(*)	0.2225	0.053	0.04	0.102	0.1
Temp range (°C)	−50 to 120	−40 to 100	−40 to 130	−5 to 125	−40 to 100	−15 to 150	−40 to 120
Best TC (ppm/°C)	3.78	5	4.6	3.98	5	3.9	11.8
Trimming	No	No	No	Yes	Yes	Yes	Yes
I_Q_ (μA)	42.13	5.2	108	38	25	38	18
PSR@27 °C	−63.1 dB @100 Hz	−28.8 dB @10 kHz	−92 dB @100 Hz	−84 dB @100 Hz	−70 dB @10 kHz	(*)	−31.8 dB @10 Hz
FOM	67.36	155.08	31.48	72.2	78.4	(*)	24

(*) Not listed.

**Table 3 micromachines-14-01420-t003:** Post layout simulations of this work.

Specification	Parameter
Process	CMOS 55 nm
Supply voltage (V)	3.3
Temp range (°C)	−50 to 120
TC (ppm/°C)	6.02
Phase margin (degree)	63.5
I_Q_ (μA)	46.8
PSR (dB)	53.6 dB@DC

## Data Availability

Not applicable.

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
