# Peer review of "A High-Precision Current-Mode Bandgap Reference with Nonlinear Temperature Compensation"

_micromachines, 2023, doi:10.3390/mi14071420_

Round 1

Reviewer 1 Report

Q1 What is a bandgap reference circuit and why is it important in electronics?

Q2 The amplifier circuit do not provide in this paper.

Q3 How does the modified equation, improve the output accuracy of the BGR circuit?

Q4 Can these solutions be applied to other types of circuits or are they specific to BGR circuits?

Q5 please provide the simulation result of output voltage (VREF) versus temperature for process corners (TT, FF, SS).

Q6 Table 2 should include the FOM

Author Response

Point 1: What is a bandgap reference circuit and why is it important in electronics?

Response 1: Because bandgap reference can generate a stable and temperature-independent voltage or current which is a standard reference for designing analog or mixed digital circuits. This module circuit is also an indispensable part of most SoC circuits.

Point 2: The amplifier circuit do not provide in this paper.

Response 2: I have provided the figure of the amplifier circuit and a simple explanation for stability after modification.

Point 3: How does the modified equation, improve the output accuracy of the BGR circuit?

Response 3: In the high-order approximation of traditional current-mode bandgap references, the differences in VEB caused by different emitter currents are ignored. This paper studies the high-order differences and modifies the equation. Make the output accuracy of the BGR circuit higher.

Point 4: Can these solutions be applied to other types of circuits or are they specific to BGR circuits? 

Response 4: They are specific to BGR circuits.

Point 5: please provide the simulation result of output voltage (VREF) versus temperature for process corners (TT, FF, SS).

Response 5: I have provided the simulation result for every process corners after modification.

Point 6: Table 2 should include the FOM

Response 6: I have designed a fair comparison parameter FOM and explained for comparison after modification.

Reviewer 2 Report

A high-precision current-mode bandgap reference (BGR) circuit with a high-order temperature compensation is presented in this paper, entitled “A High-Precision Current-Mode Bandgap Reference with Nonlinear Temperature Compensation”.

  • To achieve a high-precision BGR circuit, the equation of the nonlinear current has been modified and the high-order term of the current flowing into the nonlinear compensation bipolar junction transistor (NLCBJT) is compensated further. 
  • According to the modified equation, two solutions are designed to improve the output accuracy of BGR circuits. 
  • The first solution is to divide the NLCBJT branch into two branches to reduce the coefficient of the nonlinear temperature compensation current. 
  • The second solution is to inject the nonlinear current into the two branches respectively based on the first one to further eliminate the temperature coefficient (TC) of the current flowing into the NLCBJT. 
  • The proposed BGR circuit has been designed in Semiconductor Manufacturing International Corporation (SMIC) 55nm CMOS process. 

   In the paper, the accuracy of the BGR is improved on the basis of [8].  The rest of this paper is organized as follows: Section II describes the operation principle of the conventional current-mode BGR circuit.   Section III describes the two proposed solutions of nonlinear compensation BGR circuits. Section IV presents the simulation results that verify the accuracy of the proposed high-order terms compensated circuit. The conclusions are provided in Section V.   Figure 4 illustrates Proposed high-precision current-mode BGR circuit.   The design details of proposed high-precision current-mode BGR circuit are provided in Table I.   Figure 7 gives Layout of proposed high-precision current-mode BGR circuit.   Figure 8. Ä°s entitled as Layout of proposed high-precision current-mode BGR circuit.   I think This caption is wrong. It is stated in the text:   “With a supply voltage of 3.3V, the IREF of proposed circuit measured from -50°C to 120°C is presented in Figure 8”. Please correct Fig. 8 caption according to this.   Table 2.  gives the Performance summary and comparisons with other previous works. One important missing factor is as follows:
  • Realization circuits of the differential amplifiers A0, A1,A2 is missing. Give more details! 
  • In Fig.7 they are indicated in the layout. But they are illustrated only as symbols in the proposed circuit. Give also the realization circuits!
  I think that the paper reflects further advances in the related area. But must be improved before accepting according to the suggestions given above.
With suggested improvements the paper can be considered for publication in the journal.          

Author Response

Point 1: Realization circuits of the differential amplifiers A0, A1,A2 is missing. Give more details! 

Response 1: I have provided the circuit and description for A0, A1, A2 after modification.

Round 2

Reviewer 1 Report

The revised manuscript is recommended for publication in its current form.

Reviewer 2 Report

In this improved version of the paper entitledA High-Precision Current-Mode Bandgap Reference with Nonlinear Temperature Compensation the authors seem to have performed the suggested improvements by the reviewers. The improved sections are indicated with red colour in the text.

From my side, the amplifier circuits given in Fig.5 was very important. For this improvement the authors add the Fig.5 and necessary explanations.

I think the paper is ready now for publication.